# A multi-dimensional fusion strategy similarity measure method for patent application technology disclosure document

**Meilong Zhu[ID], Mingda Li[ID]\*, Kangwei Hou, Zhaohui Wang°, Xianjun Long°**

China Telecom Research Institute, Beijing, China

° These authors contributed equally to this work.
\* limd@chinatelecom.cn

**Data Availability Statement:** All relevant data are within the manuscript and its Supporting Information files.

## Abstract

Patent application technology disclosure document is one of the important bases for judging patent novelty and uniqueness. Automated evaluation can effectively solve the problems of long time and strong subjectivity of human evaluation. The text similarity evaluation algorithm based on corpus and deep learning technology has problems such as insufficient amount of cross-library learning data and insufficient core content tendency in the similarity judgment of patent application technology disclosure document, which limits their performance and practical application. In this paper, we propose a similarity evaluation method of patent application technology disclosure document based on multi-dimensional fusion strategy to realize the similarity measurement of patents. Firstly, in the text preprocessing section, word segmentation reconstruction and similarity evaluation optimization strategies based on word frequency and part-of-speech score weighted fusion are proposed. Then, a similarity calculation method of patent application technology disclosure document based on two new mapping spaces of dot matrix and image is proposed to achieve a more diversified comprehensive evaluation. The algorithm was evaluated by using four published text similarity matching datasets (containing 0–5 or 0/1 labels) and a set of patent application technology disclosure documents. Experimental results show that on the published text similarity matching datasets, the similarity evaluation method under the multi-dimensional fusion strategy proposed in this paper has a discrimination accuracy improvement of about 10% compared to traditional vector semantic model, and can match the discriminative ability of lightweight deep learning models without the need for training. At the same time, the discrimination accuracy of the proposed method on the sample dataset of patent application technology disclosure document is superior to traditional vector semantic model (20%) and various deep learning models (1%-8%), and the precision and recall rate are relatively balanced. The visual analysis results on the dataset of the patent application technology disclosure documents also prove the effectiveness and reliability of the similarity calculation method proposed in the dot matrix and image space, which provide a new idea and method for the similarity evaluation between patent application technology disclosure document.

**Funding:** The author(s) received no specific funding for this work.

**Competing interests:** The authors have declared that no competing interests exist.

## Introduction

The patent application technology disclosure document is a document that systematically describes the name, background, purpose, technical solution, and technical points of a patent invention by a patent applicant. It is one of the important bases for judging the novelty and uniqueness of a patent. Under the new pattern of rapid development of emerging technologies, how to improve patent quality has become one of the key issues that need to be paid attention to in the patent application process of high-tech enterprises [1]. Therefore, rapid and effective evaluation of the novelty and uniqueness of patent invention content is of great significance for patent examination.

The evaluation of patent novelty and uniqueness can be transformed into the task of text similarity evaluation under specific conditions. Nowadays, with the continuous development of artificial intelligence technology, various optimization algorithms for evaluating text similarity have also emerged in endlessly [2–5]. Xiaopeng Cao et al. added smoothing factors to the text similarity calculation, adjusted the optimal parameters through a large amount of data, and proposed a sentence scoring model to enhance semantic matching for evaluating semantic similarity [6]. Baoshan Wang designed a text similarity matching algorithm based on scoring mechanism based on word segmentation technology and dictionary vector to evaluate the similarity of various instructions [7]. Jiaqi Yang et al. proposed a method for similarity calculation of short texts based on semantic and syntactic information, which uses knowledge and corpus to express the meaning of terms to solve the polysemy problem, and uses selection parse tree to obtain the syntactic structure of short texts, thus effectively realizing the task of similarity retrieval of short texts [8]. However, compared to other language texts, the more abundant complex elements such as polysemy, synonyms, and word order logic in Chinese texts pose a more serious challenge to the robustness of traditional text similarity matching algorithms.

With the continuous development of machine learning technology, the application of text similarity matching algorithm in Chinese text has also been pushed to a new height [9–12]. Shancheng Tang et al. proposed a Chinese short text sequence similarity model based on LSTM, and used the Chinese semantic similarity data set designed by experts for training, so as to overcome the polysemy and semantic ambiguity of Chinese text to some extent [13]. Tao Lei et al. proposed a search strategy based on ElasticSearch and semantic similarity matching. Through word segmentation and preprocessing of input statements, they send them into the ER-BERT semantic similarity model for training, and finally combined with the similarity calculation formula to obtain the search results [14]. Xin Ana et al. proposed an improved method of patent similarity measurement based on entity and semantic relationship. wordNet was used as the source of lexical relationship, and Stanford, corenlp and other NLP tools were used to introduce local features based on distance and information content. Thus, the similarity between two interested patents can be better defined [15]. It is worth mentioning that although these algorithms based on deep learning can effectively improve the actual effect of text similarity matching, they all rely on specific datasets for training and iterative tuning of parameters, and still face problems to be solved, such as high difficulty in obtaining labeled datasets and long time spent in network training.

However, due to the generally long length and scattered content points of patent application technology disclosure document, which also contains many completely consistent directional descriptive prompts, it is impossible to accurately evaluate its similarity from a single sentence or paragraph compared to short sentences or specific contextual texts. Moreover, there is no publicly available dataset with similarity level annotations for training. Therefore, single use of existing traditional semantic and syntactic analysis models or deep learning models trained based on text similarity datasets are not suitable for comprehensive evaluation of

such files, and due to dataset factors, it is also impossible to effectively fine tune the deep learning model through transfer learning. Based on the above issues, this study designed a similarity evaluation method of patent application technology disclosure document based on multi-dimensional fusion strategy to realize the similarity measurement of patents. Firstly, the key parts of the patent application technology disclosure document are extracted, and the extracted text is processed through word segmentation, filtering, reconstruction, and scoring to increase the analysis weight of the content points and weaken the influence of directional description prompts. Then, the processed word list is mapped to vector, dot matrix and image space respectively to further extract more similarity features beyond semantics and syntax, and the similarity between patents is evaluated by combining the character attributes of the word itself. Finally, a more comprehensive patent similarity evaluation result is obtained by weighted and fused of the diversification evaluation results.

## Methodology

### Key text extraction

The value of each module in the technology disclosure document of patent application is evaluated, and select three modules: the invention name, background technology, and invention content and technical points for similarity evaluation. Among them, the invention content and technical points is the key evaluation object. Text extraction adopts keyword localization and region registration algorithm, that is, key fields are used to search the start and end fields of the object to be evaluated, and then the complete evaluation fields are locked through region registration. As shown in following formulas:

$$start\_F_{target_m} = [0, j+1] \; if \; Sentence \, [(i, i + \; len \, (\; target_m)), j] = \; target_m \tag{1}$$

$$end_{F_{target_m}} = [0, j-1] \; if \; Sentence \left[ \left( i, i + len\left( target_{f_m} \right) \right), j \right] = \; target_{f_m} \tag{2}$$

$$target_{area_m} = \; Sentence \, [(start_{F_{target_m}}, \; end_{F_{target_m}})] \tag{3}$$

Where $i, j$ represents the column and row coordinates of each first word in the text, $len(\cdot)$ represents the length, $target$ represents the key field, $target\_f$ represents the next field of the key field.

### Text preprocessing

Due to the fact that the similarity assessment objects in the patent application technology disclosure document are mostly short texts and are more sensitive to semantics compared to word order and sentence patterns [16]. In this study, we first use word segmentation, screening, reconstruction and scoring technologies to process the extracted evaluation fields to obtain more specific multi-level semantic information.

**Word segmentation.** Text word segmentation is based on Jieba word segmentation technology, which is suitable for Chinese and has complete technical route, simple principle and high accuracy [17]. The results of word segmentation are stored in a two-dimensional list in the form of one-to-one correspondence between words and parts of speech.

**Screening.** Words screening can be divided into three levels: symbol removal, duplicates removal and part of speech selection. At the same time, the word frequency statistics are introduced for the subsequent grading process. According to the characteristics of patent text, the key parts of speech: nouns, verbs, nominal verbs, abbreviations and idioms are screened,

which means only the words that have a great influence on semantic expression are retained, thereby improving the reliability of similarity calculation while reducing the number of calculations for each parameter.

**Reconstruction.** Since the results of word segmentation are affected by prepositions and auxiliary words in semantic expression to some extent, this paper adds a reconstruction step after the screening process, so as to eliminate the segmentation errors in the same semantic expression through the reconstruction and subdivision of the screening word list.

**Scoring.** Considering that the differences of word frequency and part of speech have more or less influence on the semantic expression and the importance of words in similarity calculation, we propose a classification mechanism for words, which is divided into part of speech priority and word frequency priority. The scoring principle is as follow:

$$word_{score_m} = k_{pos} * primary\_score - k_f * (word_{frequency_m} - 1) \tag{4}$$

Where, $k_{pos}$ represents the additional coefficient of part of speech, $k_f$ represents the additional coefficient of word frequency.

The score calculation results are stored in the scoring dictionary in the form of word (key) - score (value) as weights and introduced into the subsequent similarity calculation process to fine-tune the contribution of each word to the similarity evaluation results.

## Multi-dimensional similarity evaluation

In order to better measure the similarity of pre-processed patent application technology disclosure document, this paper improves some of the existing generic text similarity calculation methods by introducing a fine-tuning factor and scoring weights. At the same time, two new similarity evaluation schemes are proposed, which are dot matrix based similarity evaluation and image based similarity evaluation. Finally, the similarity evaluation results obtained by the above schemes are weighted and fused to obtain the multi-dimensional fusion strategy similarity evaluation results. The proposed multi-dimensional similarity evaluation method is shown in Fig 1.

**Character similarity.** In terms of the words themselves, the most intuitive similarity calculation method is the direct matching between characters. Although this calculation method is significantly weaker than the analysis and calculation of vectors for traditional text similarity calculation, it has certain advantages and significance for patent application technology disclosure documents due to its standardized wording and a large number of professional terms. In order to further improve the accuracy of character to character matching and reduce the impact of potential synonyms, we embed synonyms matching in the process of direct matching to achieve adaptive fine-tuning, so as to obtain more accurate similarity calculation results. The specific fine-tuning method is as follows:

$$str\_similarity = \frac{1}{len(wordlist)} \left( \sum_{i=0}^{len(wordlistt)} counts_i + k_s \sum_{i=0}^{len(wordlistt)} syn_i \right) \tag{5}$$

Where, $k_s = \frac{1}{\max\left(\sum_{i=0}^{len(wordlistt)} synonym_i (len(wordlist) - \sum_{i=0}^{len(wordlistt)} synonym_i)\right)}$ is the adaptive synonym matching coefficient, $counts_i$=1 if $wordlistt_i$ in wordlistc else 0 represents the result of direct character matching, and $syn_i$=1 if $wordlistt_{i\_}$ synonym in wordlistc else 0 represents the result of synonym matching.

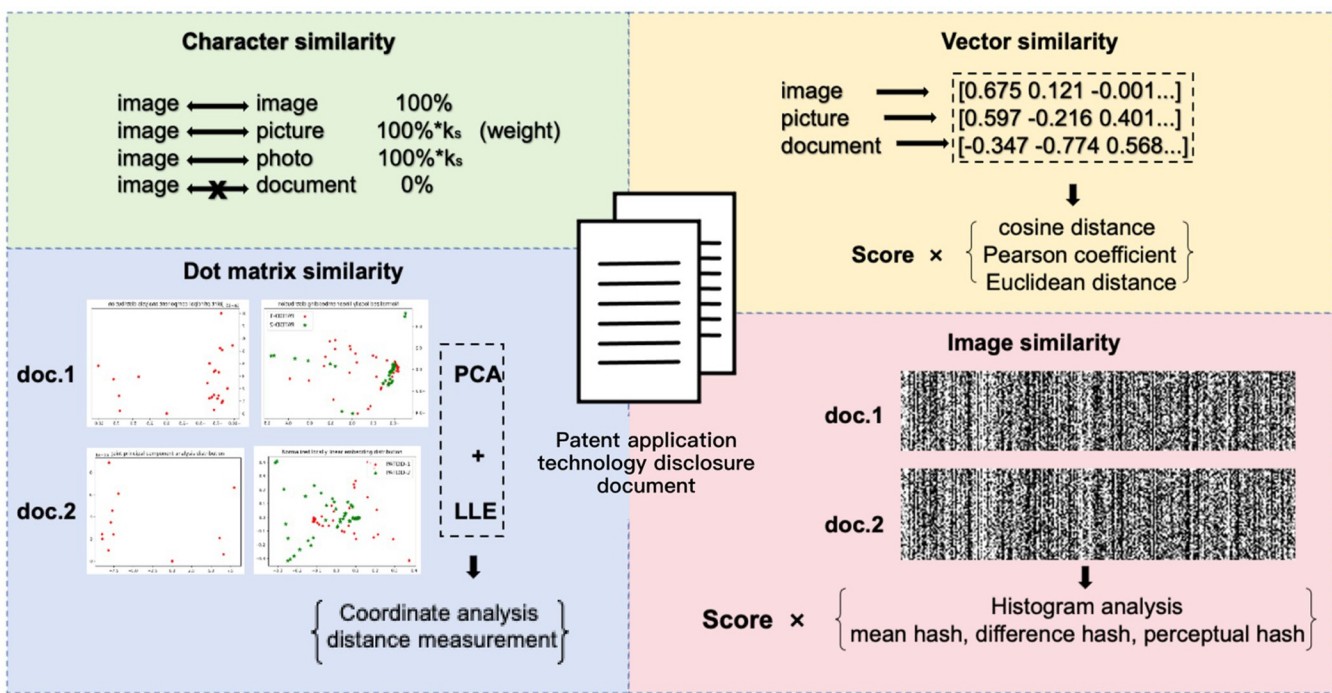

**Fig 1. The multi-dimensional similarity evaluation method.**

In addition to the optimized direct matching algorithm, we also calculate the simhash similarity and Jacquard similarity between characters to realize the comprehensive evaluation of patent application technology disclosure document character similarity.

**Vector similarity.** Word vectorization is one of the most commonly used spatial mapping methods for processing text data. In word vector space, traditional cosine distance, Pearson coefficient and Euclidean distance are the classical methods for similarity measure. On this basis, this paper introduces the scoring weight to fine-tune the measurement results, as follows:

$$optimize\_similarity_m = \ tradition\_similarity_m * \frac{dict[\ value\ ]_m}{\max(dict[\ value\ ])} \tag{6}$$

Where dict[value] represents the weight of the word vector in the scoring dictionary.

**Dot matrix similarity.** Considering that the vectorized word list belongs to multidimensional matrix, although it is difficult to be represented in multidimensional space, it can be visualized and displayed in low-dimensional space through dimensionality reduction. Principal component analysis (PCA), as a typical strategy for exploratory data analysis, feature extraction and dimension reduction, can precisely achieve this purpose [18]. Therefore, this paper proposes to apply PCA technique to the similarity evaluation between multidimensional vector matrices. It is divided into three steps:

1. Splicing each word vector matrix in the list of patent words to be evaluated with each word vector matrix in the list of comparison patent words in order to form the corresponding

two-dimensional word vector matrix, as follows:

$$M\_d\_vector_{t_i,c_i} = [\ vector_{wordlistt_i},\ vector_{wordlistc_i}]$$ (7)

2. PCA is used to reduce the dimensionality of the two-dimensional word vector matrix, and the horizontal minimum values of multiple dimensionality reduction results corresponding to each word to be evaluated are extracted as the final coordinates of the word in the dot matrix, as follows:

$$wordlistt\_co = min\left(PCA\left(M\_d\_vector_{t_i,c_k}, 2\right)\right),\ k \in (0,\ len\ (\ wordlistc\ ))$$ (8)

3. Output the dimensionality reduction results of all the words in the word list to be evaluated in the form of coordinate points, and measure the text similarity by analyzing and calculating the dot matrix diagram, as follows:

$$ave\_cal = \frac{1}{len\ (\ wordlistt\ )} \sum_{i=0}^{len(wordlistt\ )} \left(\ wordlistt\_co\_x + k_y * wordlistt_i\_co\_y\right)$$ (9)

$$PCA\_similarity = \begin{cases} 1\ ,\ ave\_cal\ = 0 \\ \alpha * ave\_cal\ + \beta,\ ave\_cal\ \neq 0 \end{cases}$$ (10)

Where, $k_y$ is the ordinate compensation factor, used to compensate the magnitude difference between the values of the horizontal and vertical coordinates, $\alpha$ and $\beta$ are the parameters.

In order to more clearly represent the spatial position relationship between each word in the word list to be evaluated and the list of words to be compared while reducing the dimension, this paper also introduces the dimension reduction method of local linear embedding (LLE). This method generates low-dimensional global coordinates by assuming that the data is located on a smooth nonlinear manifold embedded in a high-feature space [19]. The application of LLE algorithm to the visualization representation and similarity calculation of multidimensional matrix is divided into two steps:

1. Conduct n-neighborhood dimension reduction for the word vector matrix in the evaluation word list and the comparison word list respectively, and output the dimensionality reduction results in the form of coordinate points as a dot matrix diagram and a two-dimensional list respectively.

2. Calculate the minimum Euclidean distance between the coordinates of each point in the two-dimensional list generated from the word list to be evaluated and the coordinates of each point in the two-dimensional list generated from the comparison word list respectively. The final similarity evaluation result is obtained by accumulating and averaging the minimum values obtained by parameter operation. As shown in the following formulas:

$$distance_i = \min(\sqrt{(y_{t_i} - y_{c_k})^2 + (x_{t_i} - x_{c_k})^2}), k \in (0, len(wordlistc))$$ (11)

$$LLE\_similarity = \begin{cases} 1 \, , max([\,distance\,]) = 0 \\ \dfrac{\sum_{i=0}^{len(wordlistt)} distance_i}{len(\,wordlistt)*max([\,distance\,])} \, , max([\,distance\,]) \neq 0 \end{cases} \qquad (12)$$

**Image similarity.** From another perspective, we can regard the multidimensional matrix after word vectorization as a special single channel image matrix, so this paper proposes that we can further evaluate the similarity of patent application technology disclosure document from an image perspective. In the field of image, the histogram of an image is one of the important ways to display the statistical characteristics of the image, in other words, it can be considered as an approximation of the grayscale density function of the image [20]. The similarity evaluation of vectorized multidimensional matrix with histogram is divided into the following three steps:

1. Normalize the vectorized multidimensional matrix and reduce the numerical value of each dimension to the range of [0, 255], as follows:

$$wordlist_{i_{j}\_}new = \begin{cases} min\left(\dfrac{wordlist_{i_j}}{ave(\,wordlist_i)}*255, 255\right), \; wordlist_{i_j} \geq 0 \\ 255 + max\left(\dfrac{wordlist_{i_j}}{ave(\,wordlist_i)}*255, -255\right), \; wordlist_{i_j} < 0 \end{cases} \qquad (13)$$

2. Draw the histogram of the normalized multidimensional matrix and calculate the matching degree. The specific calculation method is shown in the following formula:

$$degree_i = \sum_{j=0, k=0}^{len(\,wordlistc),\; len(wordlistt_i)} max\left(\dfrac{min(\,histogramt_{i_k}, \; histogramc_{j_k})}{max(\,histogramt_{i_k}, \; histogramc_{j_k})}\right) \qquad (14)$$

3. Weight and sum the histogram matching calculation results generated by each word vector, and then the average is taken as the final similarity measurement result, as shown in the following formula:

$$hist_{similarity} = \frac{1}{len(wordlistt)} \sum_{i=0}^{len(wordlistt)} degree_i * \frac{dict[\,value\,]_i}{max(dict[\,value\,])} \qquad (15)$$

At the same time, we also directly use the mean hash, difference hash and perceptual hash algorithms to calculate the similarity degree of the normalized single-channel images [21–23],

and construct a set of similarity conversion functions to evaluate the fixed interval, as follows:

$$\overline{MH} = \frac{1}{3}\left(abs(\,ahash_t - ahash_c) + abs(\,dhash_t - \,dhash_c) + abs(\,phash_t - \,phash_c)\right) \quad (16)$$

$$Mhash\_similarity = \begin{cases} \dfrac{1}{100}\left(\alpha*\overline{MH}^2 - \beta*\overline{MH} + \gamma\right), \overline{MH} \le \mu \\[2mm] \dfrac{1}{100}\left(\omega*\overline{MH} + \varphi\right), \overline{MH} > \mu \end{cases} \quad (17)$$

Where, $\overline{MH}$ represents the average hash distance, *ahash* represents the average hash, *dhash* represents the difference hash, *phash* represents the perceptual hash, $\mu$ represents the similarity conversion function threshold and α、β、γ、ω、φ are the parameters.

**Weighted fusion.**  Finally, ten similarity calculation results obtained by four similarity measurement methods are weighted and fused to achieve multi-dimensional similarity evaluation of patent application technology disclosure document, as follows:

*Multivariate similarity*

$$= \frac{1}{10}(\alpha(hist\_similarity \,+\, pearson\_similarity \,+\, cosine\_similarity + \, jaccard\_similarity$$

$$+ \, simhash\,similarity\,) + \beta(ED\_similarity + str\_similarity \,+ LLE\_similarity$$

$$+PCA\_similarity + Mhash\_similarity) \quad (18)$$

Where *pearson_similarity* represents Pearson similarity, *cosine_similarity* represents cosine similarity, *jaccard_similarity* represents Jacquard similarity, *simhash_similarity* represents sim hash similarity, *ED_similarity* represents Euclidean distance similarity, $\alpha$ and $\beta$ are the parameters.

## Results and discussion

### Data samples

Five datasets were used to evaluate the patent similarity evaluation method under the multi-fusion strategy proposed in this paper.

A.  A total of 8050 pairs of corpora from the Chinese-STS-B dataset [24], including 6 labels ranging from 0 to 5. The larger the number, the more similar the text pairs are.

B.  A total of 260068 pairs of corpora from the Chinese question-and-answer matching Lcqmc dataset published by HIT [25], including 0 (dissimilar) and 1 (similar) labels.

C.  A total of 38,650 pairs of corpora from the AFQMC Ant Financial semantic similarity dataset [26], including 0 (dissimilar) and 1 (similar) labels.

D.  A total of 320 pairs of corpus from the small cloth assistant dialogue short text matches bustm dataset(from Chinese small sample learning evaluation dataset FewCLUE) [27], including 0 (dissimilar) and 1 (similar) labels.

E.  A total of 8000 patent application technology disclosure documents from China Telecom and publicly available online.

## Comparison with state-of-the-art models

Two general evaluation datasets (A, B) from the above five datasets were used to compare the results of the proposed algorithm with the existing advanced similarity evaluation models. Vector and text semantic model, BiLSTM+self attention [28], DiSAN [29], RoBERTa-wwm-ext [30], XLNet-mid [31], BERT [32] and the proposed algorithm were tested on the dev and test datasets of Chinese STS-B [24] and LCQMC [25] as shown in Table 1. The confusion matrices of the evaluation results obtained by using the multi-dimensional fusion strategy are shown in Fig 2.

From the comparative experimental results of the general evaluation dataset in Table 1, it can be seen that the similarity evaluation algorithm proposed in this paper has a much better evaluation ability on the Chinese text matching dataset than traditional vector and text semantic models, slightly higher than lightweight deep learning models. However, there are still some gaps in the evaluation results compared to heavyweight semantic models such as BERT. This indicates that the multi-dimensional fusion strategy algorithm proposed in this article is more effective and comprehensive in extracting text features compared to traditional vector semantic models. In addition, compared to deep learning models, it can comprehensively evaluate the similarity of any text without training., thus effectively solve the problems of high difficulty in obtaining training sets for deep learning models and long network training ticme.

It can be seen from the confusion matrices in Fig 2 that the number of samples with correct discrimination is higher than the number of samples with wrong discrimination in each similarity level except level 1 in the Chinese-STS-B dataset [24]. Among them, samples with similarity level of 0 and 3 have the highest discrimination accuracy, but there is a certain degree of cross misjudgment between level 1 & 2 as well as level 4 & 5. This may be caused by the blurred boundaries of similarity between these levels.

## Validation of competition dataset

We also used two competition datasets (C, D) from the above five datasets to evaluate the results of the proposed algorithm and compared them with the best evaluation results generated by the competition under the same process.

It can be observed from Table 2 that the multi-dimensional fusion strategy proposed in this article can also approach the optimal discrimination results of heavyweight deep learning large models on the competition dataset without the need of training. Further proves the effectiveness and generalization of the method proposed in this paper in the task of text similarity discrimination.

**Table 1. Test result comparison of general datasets.**

| Model | Chinese-STS-B [24] | | Model | LCQMC [25] | |
|---|---|---|---|---|---|
| | Dev | Test | | Dev | Test |
| Vector and text semantic | 32.10 | 29.99 | Vector and text semantic | 58.17 | 61.47 |
| BiLSTM+self-attention [28] | 43.87 | 41.24 | RoBERTa-wwm-ext [30] | 82.98 | 82.28 |
| DiSAN [29] | 44.21 | 42.09 | XLNet-mid [31] | 82.00 | 84.00 |
| BERT [32] | 53.84 | 50.26 | BERT [32] | 81.29 | 82.70 |
| **multi-dimensional fusion strategy(ours)** | **45.75** | **42.27** | **multi-dimensional fusion strategy(ours)** | **73.19** | **76.38** |

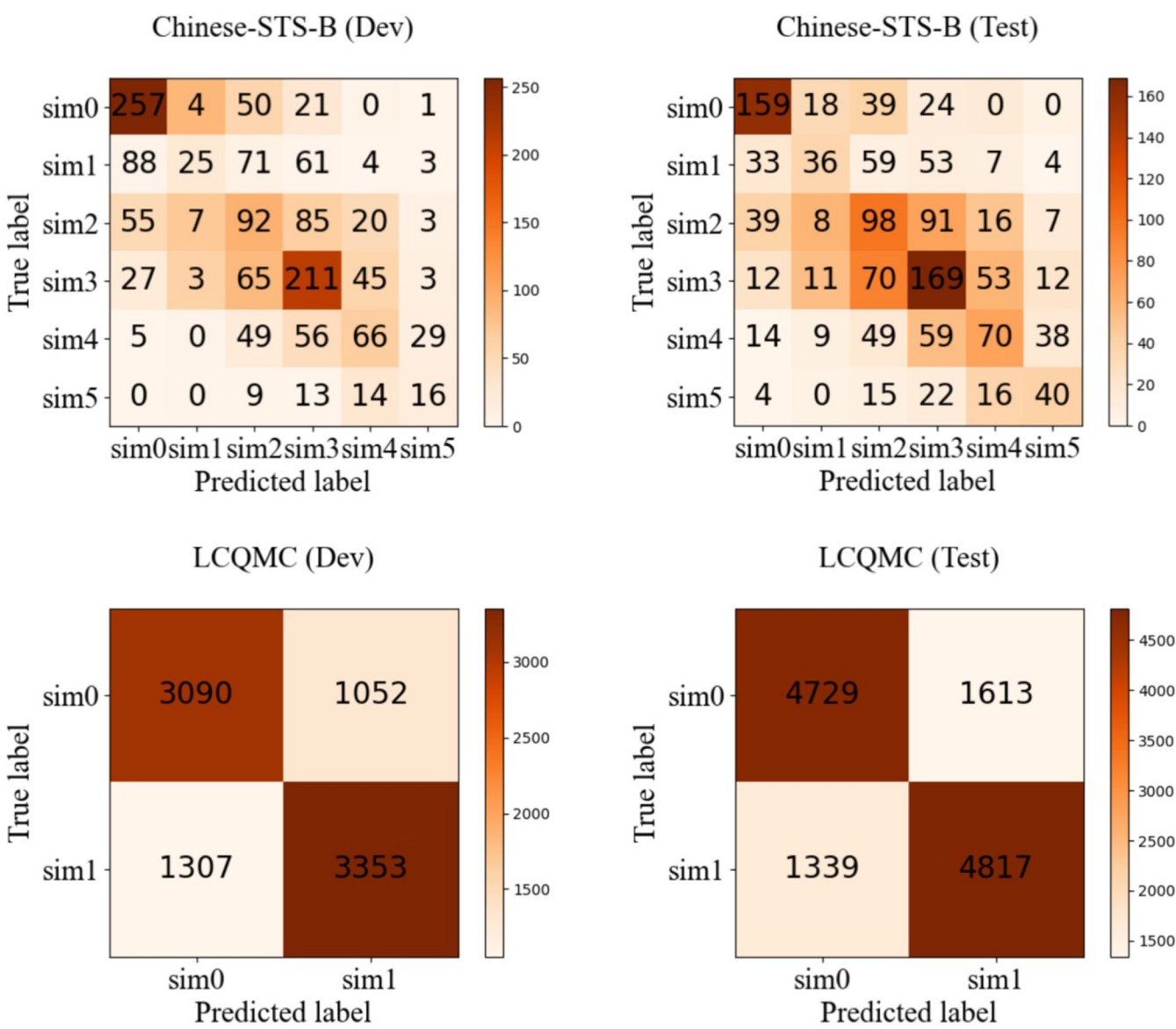

**Fig 2. The confusion matrices of the evaluation results obtained using multi-dimensional fusion strategy.**

### Verification of patent application technology disclosure document

The parameters of the proposed patent application technology disclosure document similarity evaluation method based on multi-dimensional fusion strategy were set empirically and are shown in Table 3. The comprehensive evaluation ability of the proposed algorithm were tested

**Table 2. Evaluation result of competition datasets.**

| Database | Evaluation item | | | |
|---|---|---|---|---|
| | **Train** | **Dev** | **Test** | **Test(best)** |
| AFQMC [26] | 72.41 | 70.13 | 69.88 | 74.44 (RoBERTa-wwm-large) |
| FewCLUE (bustm) [27] | 70.63 | 68.13 | 66.14 | 72.42 (RoBERTa-large) |

**Table 3. Parameters selected for multi-dimensional fusion patent application technology disclosure document similarity evaluation.**

| | |
|---|---|
| PCA vertical coordinate compensation factor ($k_y$) | 1e15 |
| PCA similarity measurement parameters ($\alpha$、$\beta$) | (-0.05, 0.975)、(-0.088, 1.106)、(-0.085, 1.1) |
| Number of LLE dimensionality reducing neighborhoods (n) | 3 |
| Multiple Hash Similarity Measurement Parameters ($\alpha$、$\beta$、$\gamma$、$\omega$、$\varphi$) | (3/25, 4.2, 95, -10/3, 265/3) |
| Similarity conversion function threshold ($\mu$) | 12 |
| multi-dimensional similarity weighted integration parameters ($\alpha$、$\beta$) | (2/3, 4/3) |

by using the sample dataset of patent application technology disclosure document. Firstly, we conducted ablation experiments to verify the effectiveness of character matching in similarity evaluation in patent application technology disclosure document, the test results are shown in Table 4. Then, the effectiveness of the word segmentation reconstruction and word frequency and part-of-speech score weighted introduced in this paper were verified in eliminating semantic expression errors and refining word frequency and part of speech contributions, as shown in Fig 3 and Table 5. After that, two representative texts with comprehensive similarity evaluation results below 0.5 and above 0.5 were selected for visualization analysis and discussion. The visualization results in the dot matrix and image space obtained by taking the invention name and invention content and technical points as examples are shown in Figs 4 and 5. Finally, we manually selected and labeled 210 pairs of data from the 8000 patent application technology disclosure documents (E) in a ratio of approximately 1:1 between similar and non similar samples to form a test dataset to further compare and evaluate the effectiveness of the algorithm proposed in this paper, as shown in Table 6.

It can be seen from Table 4 that in the similarity evaluation task of patent application technology disclosure documents, the introduction of similarity calculation results between characters can effectively improve the comprehensive evaluation performance of the algorithm. This further shows that for long texts with standardized wording and more professional terms, the similarity between characters can indeed effectively highlight the similarity between texts to a certain extent like the similarity between vectors.

It can be observed from Fig 3 that compared with the simple use of filtered word segmentation results, the reconstructed results effectively eliminate the influence of prepositions and auxiliary words on word segmentation results in semantic expression, which is more conducive to the subsequent determination of similarity between word vectors. And the ablation test in Table 5 also confirms the effectiveness of the reconstruction and scoring sections in the final similarity evaluation.

As shown in the visualization results of the dot matrix space in Fig 4, the visualization results of joint principal component analysis indicate that texts with higher similarity generate fewer discrete non overlapping points in the dot matrix space, and the number of significant clustering of points is also reduced. This is basically consistent with the discriminant principle of PCA in similarity analysis of different samples, that is, high similarity between clustered

**Table 4. Ablation test results for character matching.**

| Models | Accuracy | Precision | Recall | F1 score |
|---|---|---|---|---|
| multi-dimensional fusion strategy without character matching | 0.7048 | 0.7054 | 0.7315 | 0.7182 |
| multi-dimensional fusion strategy with character matching | 0.8048 | 0.7863 | 0.8519 | 0.8178 |

```
[ 'sample' ,  'image' ,  'identification method' ,  'device' ,  'equipment' ,  readable ,  'storage medium' ]

[ 'scene' ,  'image identification' ,  'method' ,  'device' ,  'electronic equipment' ,  'computer' ,  'able' ,  'read' ,  'medium' ]
```

Word segmentation results before reconstruction

```
[ 'sample' ,  'image identification' ,  'method' ,  'device' ,  'equipment' ,  'able' ,  'read',  'storage medium' ]

[ 'scene' ,  'image identification' ,  'method' ,  'device' ,  'electronic equipment' ,  'computer' ,  'able' ,  'read' ,  'medium' ]
```

Word segmentation results after reconstruction

**Fig 3. Reconstruction result.**

sample points and low similarity between dispersed sample points. Meanwhile, the visualization results of normalized local linear embedding indicate that the higher the similarity, the stronger the linear relationship between the various communities of generated points in the dot matrix space, the higher the similarity of function fitting, and the smaller the sum of the minimum distances between all points contained in each community. This is also basically consistent with the dimensionality reduction principle of local linear embedding, that is, the local structure of data is kept unchanged, and the relationship between k nearest neighbors is constrained by loss function. From these, it can be seen that the dot matrix similarity evaluation principle proposed in this article can effectively demonstrate the similarity between texts.

From the visualization results of image space shown in Fig 5, it can be seen that the higher the similarity, the closer the distribution of grayscale values in the image space of the text, and the weaker the step property of horizontal and vertical pixel points. This is consistent with the histogram distribution pattern of similar images, indicating that the image matrix generated by text vectors can also effectively demonstrate the similarity between texts.

It can be seen from the comparative experimental results in Table 6 that the multi-dimensional fusion strategy similarity measure method proposed in this paper has much higher evaluation capabilities than general vector semantic and word frequency models in the sample data set of patent application technology disclosure documents, and has certain advantages in accuracy and precision rate compared with various trained deep learning models and their optimization models. Although the recall rate is relatively low, the recall and precision rate are relatively balanced under the premise that the number of positive and negative samples is basically the same. This shows that the algorithm proposed in this paper has stronger feature extraction ability and more comprehensive evaluation ability in the similarity evaluation task of patent application technology disclosure documents.

**Table 5. Ablation test results for reconstruction and scoring sections.**

| Models | Accuracy | Precision | Recall | F1 score |
|---|---|---|---|---|
| multi-dimensional fusion strategy without reconstruction and scoring sections | 0.6810 | 0.6242 | 0.9537 | 0.7476 |
| multi-dimensional fusion strategy with scoring section | 0.7238 | 0.7083 | 0.7870 | 0.7456 |
| multi-dimensional fusion strategy with reconstruction section | 0.7000 | 0.6471 | 0.9167 | 0.7586 |
| multi-dimensional fusion strategy with reconstruction and scoring sections | 0.8048 | 0.7863 | 0.8519 | 0.8178 |

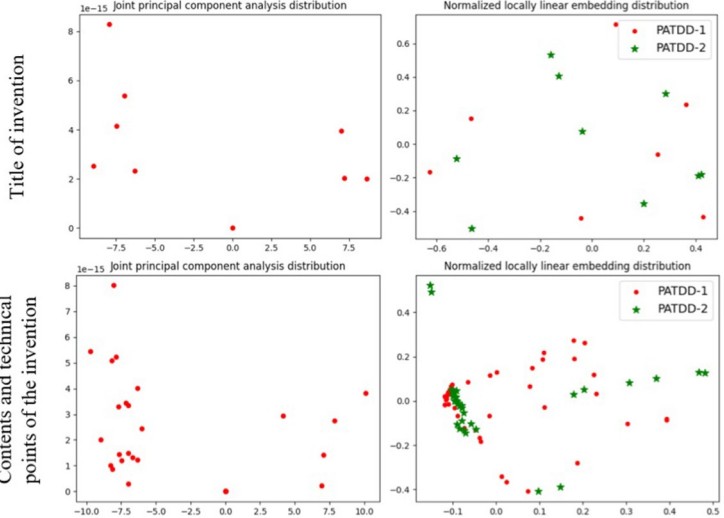

Multi-dimensional similarity evaluation value: 0.4756

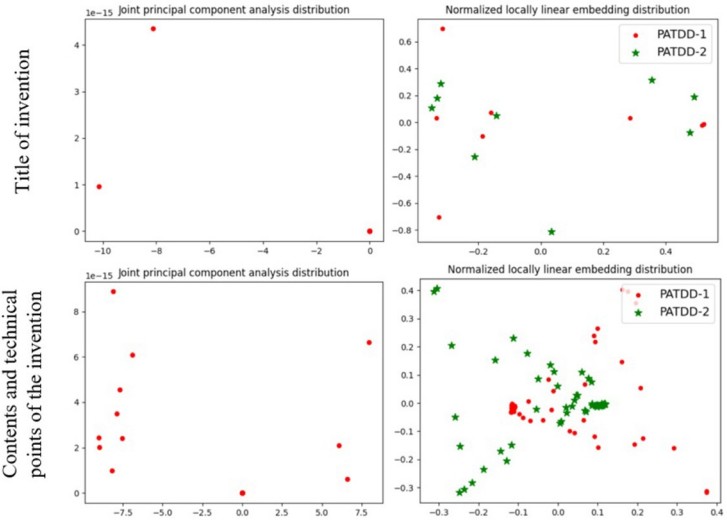

Multi-dimensional similarity evaluation value: 0.7529

**Fig 4. Dot matrix space visualization results.**

## Conclusions

The technology disclosure document of patent application is an important basis for judging the novelty and uniqueness of a patent. However, due to the diverse logical expression of patented text, large number of proper nouns, and the difficulty of obtaining labeled datasets, traditional vector semantic models usually cannot obtain satisfactory discriminant results, and the transfer learning ability of deep learning models is also greatly limited. In this study, by

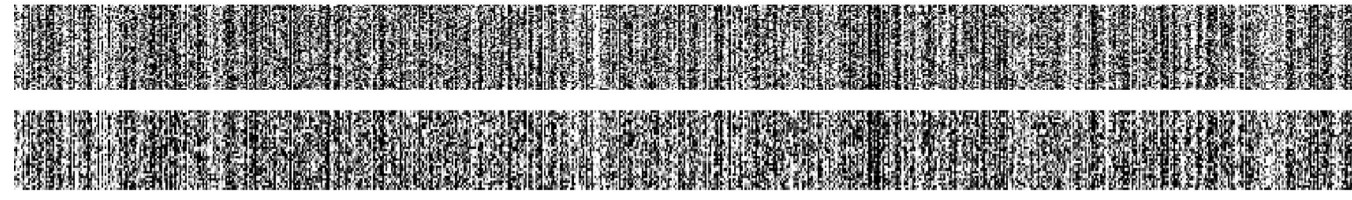

Multi-dimensional similarity evaluation value：0.4756

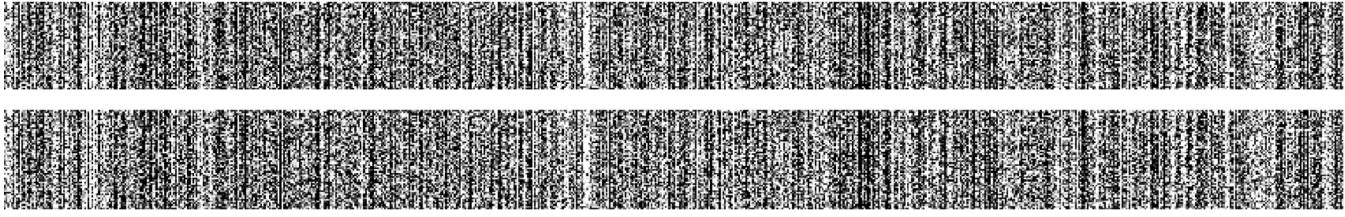

Multi-dimensional similarity evaluation value：0.7529

**Fig 5. Image space visualization results.**

multi-dimensional spatial mapping of the preprocessed word list, combined with the word frequency and part of speech of words in the sentence, a similarity assessment method of patent application technology disclosure document under the multi-dimensional fusion strategy is proposed to achieve the measurement of patent similarity.

The experimental results show that: 1) The reconstruction section effectively eliminate the influence of prepositions and auxiliary words on word segmentation results in semantic expression, and the scoring section effectively improves the contribution of different parts of speech and word frequency in the similarity assessment; 2) The similarity evaluation method under the multi-dimensional fusion strategy is better than the traditional algorithm in the general text similarity matching dataset, which is comparable to the lightweight deep learning model and slightly inferior to the heavyweight deep learning model; 3) The introduction of the similarity assessment method of dot matrix and image space can effectively highlight the similarity between patent application technology disclosure documents; 4) The multi-dimensional fusion strategy similarity measure method has a better evaluation effect than the traditional algorithm and also has certain advantages compared with the deep learning model in the sample dataset of patent application technology disclosure documents.

**Table 6. Evaluation result of the sample dataset of patent application technology disclosure documents.**

| Models | Accuracy | Precision | Recall | F1 score |
|---|---|---|---|---|
| Vector and text semantic | 0.6048 | 0.5698 | 0.9444 | 0.7108 |
| TF-IDF [33] | 0.6381 | 0.9211 | 0.3241 | 0.4795 |
| BERT [32] | 0.7238 | 0.6736 | 0.8981 | 0.7698 |
| S-BERT [34] | 0.7857 | 0.7086 | 0.9907 | 0.8263 |
| XLNet-mid [31] | 0.7952 | 0.7338 | 0.9444 | 0.8259 |
| **multi-dimensional fusion strategy(ours)** | 0.8048 | 0.7863 | 0.8519 | 0.8178 |

Although the proposed method is better than other methods in the evaluation results of the sample dataset of patent application technology disclosure documents, there are also some shortcomings at the method and data level. Therefore, further research can be carried out from the following directions: 1) For the positive and negative samples in the algorithm evaluation results, various statistical analysis methods can be used to dig deeper into the intra-class consistency and inter-class differences, so as to further optimize the parameters of the proposed algorithm; 2) A larger and more accurate dataset for evaluating the similarity of patent application technology disclosure documents can be constructed to evaluate the performance of the algorithm more comprehensively; 3) The multi-dimensional fusion strategy can be combined with the feature extraction module of the deep learning model to improve the diversified feature extraction ability of the deep learning model, so as to further enhance the similarity evaluation performance of various deep learning models.

## Supporting information

**S1 Dataset. Chinese-STS-B dataset.**
(TXT)

**S2 Dataset. Lcqmc dataset.**
(TSV)

**S3 Dataset. AFQMC ant financial semantic similarity dataset.**
(TXT)

**S4 Dataset. Bustm dataset.**
(TXT)

**S5 Dataset. Patent application technical disclosure documents dataset (A).**
(XLSX)

**S6 Dataset. Patent application technical disclosure documents dataset (B).**
(XLSX)

**S1 File. Sample of patent application technical disclosure documents.**
(TXT)

## Acknowledgments

The authors would like to thank China Telecom for providing the data of the patent application technology disclosure documents.

## Author Contributions

**Conceptualization:** Meilong Zhu, Mingda Li, Kangwei Hou, Zhaohui Wang.

**Data curation:** Kangwei Hou.

**Formal analysis:** Meilong Zhu.

**Funding acquisition:** Xianjun Long.

**Methodology:** Meilong Zhu.

**Project administration:** Mingda Li, Zhaohui Wang.

**Resources:** Xianjun Long.

**Supervision:** Zhaohui Wang.

**Validation:** Xianjun Long.

**Visualization:** Meilong Zhu.

**Writing – original draft:** Meilong Zhu.

**Writing – review & editing:** Mingda Li.

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
