## [Decision Letter · Decision Letter 0]

26 Jul 2023

PONE-D-23-12263A multi-dimensional fusion strategy similarity measure method for patent application technology disclosure documentPLOS ONE

Dear Dr. Li,

Thank you for submitting your manuscript to PLOS ONE. After careful consideration, we feel that it has merit but does not fully meet PLOS ONE’s publication criteria as it currently stands. Therefore, we invite you to submit a revised version of the manuscript that addresses the points raised during the review process. Please submit your revised manuscript by** **Sep 09 2023 11:59PM.  If you will need more time than this to complete your revisions, please reply to this message or contact the journal office at plosone@plos.org. Please include the following items when submitting your revised manuscript:A rebuttal letter that responds to each point raised by the academic editor and reviewer(s). You should upload this letter as a separate file labeled 'Response to Reviewers'.A marked-up copy of your manuscript that highlights changes made to the original version. You should upload this as a separate file labeled 'Revised Manuscript with Track Changes'.An unmarked version of your revised paper without tracked changes. You should upload this as a separate file labeled 'Manuscript'.

We look forward to receiving your revised manuscript.

Kind regards,

Anas Bilal, Ph.D.

Academic Editor

PLOS ONE

Journal requirements

Reviewers' comments:

Reviewer's Responses to Questions

**Comments to the Author**

1. Is the manuscript technically sound, and do the data support the conclusions?

Reviewer #1: Partly

Reviewer #2: Yes

2. Has the statistical analysis been performed appropriately and rigorously? 

Reviewer #1: Yes

Reviewer #2: Yes

3. Have the authors made all data underlying the findings in their manuscript fully available?

Reviewer #1: No

Reviewer #2: Yes

4. Is the manuscript presented in an intelligible fashion and written in standard English?

Reviewer #1: Yes

Reviewer #2: Yes

5. Review Comments to the Author

Reviewer #1: This paper proposes a multi-dimensional fusion strategy similarity measure method for patent application technology disclosure documents by using many machine-learning-based similarity measurement approaches. The paper targets an important topic, but there are several problems with this paper.

1.The introduction does not discuss the specific challenges faced by the task of measuring the disclosure documents, which is very important for your paper. First, we cannot know why the existing deep learning-based text similarity analysis cannot solve the current task. Second, we cannot know what motivated you to design your model, and why your model is effective.

2.The model considers character similarity, which has been shown by many papers to be weaker than word vector analysis, so what is the point of its existence, and has it been followed up with ablation experiments to prove it?

3.All of the analysis approaches are based on machine learning or simple statistical analysis; I cannot find any innovations in modeling.

4.Most of the datasets used in the experiment are designed for common text analysis. Tests against these datasets are not meaningful for the scenarios in this paper, and the proposed method is significantly weaker than deep-learning-based methods. And, the only effective dataset is not open-source. I think the patent application technology disclosure documents have already been released after they are submitted, so why not open source these documents?

5.Without doing comparison experiments on the only valid dataset, we cannot know the superiority of the proposed method over existing text similarity analysis methods.

Reviewer #2: - The proposed similarity assessment method of patent application technical disclosure documents under a multi-dimensional fusion strategy is novel.

- The proposed method significantly improves the discriminant accuracy by about 10% compared with the traditional vector semantic model.

- The proposed method is evaluated using four published text similarity matching datasets (containing 0-5 or 0/1 labels) and a set of patent application technology disclosure documents.

- It would be nice to add an overview figure to the paper to represent the proposed multidimensional fusion strategy.

- The proposed method can be compared to the discrimination ability of a lightweight deep learning model without training. However, the deep learning model outperforms the proposed method by a large margin. Therefore, this paper is limited in terms of usability. Furthermore, it is expected that higher performance can be achieved by applying multi-dimensional fusion strategies to deep learning models.

- It may be difficult for readers without a background in the field to understand, so it is necessary to explain in detail the characteristics of patent documents and multidimensional fusion strategies.

- The paper does not provide a detailed discussion of the limitations of the proposed method and areas for future research.

XLnet → XLNet

、→ ,

6. PLOS authors have the option to publish the peer review history of their article (what does this mean?). If published, this will include your full peer review and any attached files.

Reviewer #1: No

Reviewer #2: **Yes: **Suan Lee

---

## [Author Response · Author response to Decision Letter 0]

14 Aug 2023

Dear academic editor and reviewers:

Thanks for the careful review of our manuscript. Your recognition and suggestions are crucial for us. We have revised the manuscript carefully, and please find the following detailed reply and actions to your comments and suggestions one by one:

Academic editor:

1.Please ensure that your manuscript meets PLOS ONE's style requirements, including those for file naming. The PLOS ONE style templates can be found at https://journals.plos.org/plosone /s/file?id=wjVg/PLOSOne_formatting_sample_main_body.pdf and https://journals.plos.org/ploso ne/s/file?id=ba62/PLOSOne_formatting_sample_title_authors_affiliations.pdf.

Reply:Thanks for this reminder, we do have some problems with the format of the manuscript we submitted before.

Action:We have carefully read the submission specification of PLOS One journal and the pdf file you provided, and systematically revised and improved our manuscript according to the requirements therein.

2.In your Data Availability statement, you have not specified where the minimal data set underlying the results described in your manuscript can be found. PLOS defines a study's minimal data set as the underlying data used to reach the conclusions drawn in the manuscript and any additional data required to replicate the reported study findings in their entirety. All PLOS journals require that the minimal data set be made fully available.

Reply:Thanks for this comment. We have uploaded part of the supporting dataset of the experimental results of this paper in the supplementary materials when we submitted our manuscript before. The remaining datasets will be supplemented during the upload process for our revised manuscript.

Action:We have supplemented the smallest datasets previously uploaded, and we have added a section of the data availability statement in our paper. The latest datasets will be able to be used to draw all the conclusions in our paper.

Reviewer #1: This paper proposes a multi-dimensional fusion strategy similarity measure method for patent application technology disclosure documents by using many machine-learning-based similarity measurement approaches. The paper targets an important topic, but there are several problems with this paper.

Reply:Thank you for your professional review of our manuscript. As you are concerned, there are several problems that need to be addressed. According to your precious suggestions, we have revised the manuscript carefully, the detailed reply and actions are listed below:

1.The introduction does not discuss the specific challenges faced by the task of measuring the disclosure documents, which is very important for your paper. First, we cannot know why the existing deep learning-based text similarity analysis cannot solve the current task. Second, we cannot know what motivated you to design your model, and why your model is effective.

Reply:Thanks for this comment, it should be discussed specifically in the introduction part.

Action:We have added an analysis of the characteristics of the patent application technology disclosure documents in the introduction section of the article. Based on the analysis of features, the problems and challenges of existing methods in solving the similarity measurement of the above documents are discussed, and supplementary experiments are added in subsequent chapters to prove the effectiveness of the proposed algorithm.

2.The model considers character similarity, which has been shown by many papers to be weaker than word vector analysis, so what is the point of its existence, and has it been followed up with ablation experiments to prove it?

Reply:Thanks for your question in this section. It is true that in most cases, the analysis of character similarity is indeed weaker than that of word vectors, but due to the unique characteristics of patent application technical disclosure documents, character similarity can play a positive role in the overall measurement of similarity of the above documents.

Action:We have added an analysis and discussion of the above issues in the module on character similarity.

3.All of the analysis approaches are based on machine learning or simple statistical analysis; I cannot find any innovations in modeling.

Reply:Thank you for pointing this out. Although we have not made structural innovations for related models, we have put forward new insights in the perspective and method of text similarity analysis, such as optimizing the computational bias of similarity by introducing reconstruction and scoring sections, mapping text vectors to dot matrix and image spaces to analyze and calculate their similarity, etc. And we are planning to try to combine the proposed method with the feature extraction module of the deep learning model to achieve better performance.

4.Most of the datasets used in the experiment are designed for common text analysis. Tests against these datasets are not meaningful for the scenarios in this paper, and the proposed method is significantly weaker than deep-learning-based methods. And, the only effective dataset is not open-source. I think the patent application technology disclosure documents have already been released after they are submitted, so why not open source these documents?

Reply:Thanks for this comment. Our experiments on the common datasets are intended to demonstrate the feasibility of the proposed method in the similarity evaluation tasks, although there is a certain gap in performance compared to deep learning models, it also has certain advantages. As for the problem of patent application technology disclosure document dataset, we have obtained an open source license for it.

Action:We have collated all remaining datasets related to the output of the results of the experiments in the paper and submitted them as supplementary material along with the revised manuscript.

5.Without doing comparison experiments on the only valid dataset, we cannot know the superiority of the proposed method over existing text similarity analysis methods.

Reply:Thank you for pointing this out. This is indeed an indispensable part of the experiments to prove the superiority of the proposed method in our paper.

Action:We have supplemented this part of the experiment and discussed it based on the results of the comparative experiment.

Reviewer #2: The proposed similarity assessment method of patent application technical disclosure documents under a multi-dimensional fusion strategy is novel. The proposed method significantly improves the discriminant accuracy by about 10% compared with the traditional vector semantic model. The proposed method is evaluated using four published text similarity matching datasets (containing 0-5 or 0/1 labels) and a set of patent application technology disclosure documents.

Reply:Thanks for the careful review of our manuscript. We sincerely appreciate your recognition and encouragement. We have revised the manuscript carefully, the detailed reply and actions are listed below:

1.It would be nice to add an overview figure to the paper to represent the proposed multidimensional fusion strategy.

Reply:Thanks for this suggestion, we accept it thoroughly.

Action: We have added an overview figure of the multi-dimensional fusion strategy similarity measure method proposed in this paper in the methodology section of the manuscript.

2.The proposed method can be compared to the discrimination ability of a lightweight deep learning model without training. However, the deep learning model outperforms the proposed method by a large margin. Therefore, this paper is limited in terms of usability. Furthermore, it is expected that higher performance can be achieved by applying multi-dimensional fusion strategies to deep learning models.

Reply:Thank you for pointing this out. The evaluation performance of our proposed method on the common datasets of text similarity does have a certain gap with that of the deep learning models, but we have added a supplementary experiment to confirm that the proposed method has strong advantages for the similarity evaluation task of patent application technology disclosure document data. And we are planning to try to combine the proposed method with the feature extraction module of the deep learning model according to your suggestion in the follow-up study to achieve better performance.

3.It may be difficult for readers without a background in the field to understand, so it is necessary to explain in detail the characteristics of patent documents and multidimensional fusion strategies.

Reply:Thanks for this suggestion, it is indeed necessary.

Action:We have added a detailed description of the characteristics of patent application technology disclosure document and a brief supplement to the multi-dimensional strategy in the introduction of the manuscript.

4.The paper does not provide a detailed discussion of the limitations of the proposed method and areas for future research.

Reply:Thank you for pointing this out. We did ignore this part.

Action:We have added a discussion of the limitations of the proposed method and an outlook for future research directions at the end of the conclusions section.

---

## [Decision Letter · Decision Letter 1]

3 Sep 2023

PONE-D-23-12263R1A multi-dimensional fusion strategy similarity measure method for patent application technology disclosure documentPLOS ONE

Dear Dr. Li,

Thank you for submitting your manuscript to PLOS ONE. After careful consideration, we feel that it has merit but does not fully meet PLOS ONE’s publication criteria as it currently stands. Therefore, we invite you to submit a revised version of the manuscript that addresses the points raised during the review process. Please submit your revised manuscript by Oct 18 2023 11:59PM. If you will need more time than this to complete your revisions, please reply to this message or contact the journal office at plosone@plos.org. Please include the following items when submitting your revised manuscript:A rebuttal letter that responds to each point raised by the academic editor and reviewer(s). You should upload this letter as a separate file labeled 'Response to Reviewers'.A marked-up copy of your manuscript that highlights changes made to the original version. You should upload this as a separate file labeled 'Revised Manuscript with Track Changes'.An unmarked version of your revised paper without tracked changes. You should upload this as a separate file labeled 'Manuscript'.

We look forward to receiving your revised manuscript.

Kind regards,

Anas Bilal, Ph.D.

Academic Editor

PLOS ONE

Reviewers' comments:

Reviewer's Responses to Questions

**Comments to the Author**

1. If the authors have adequately addressed your comments raised in a previous round of review and you feel that this manuscript is now acceptable for publication, you may indicate that here to bypass the “Comments to the Author” section, enter your conflict of interest statement in the “Confidential to Editor” section, and submit your "Accept" recommendation.

Reviewer #1: (No Response)

Reviewer #2: All comments have been addressed

2. Is the manuscript technically sound, and do the data support the conclusions?

Reviewer #1: (No Response)

Reviewer #2: Yes

3. Has the statistical analysis been performed appropriately and rigorously? 

Reviewer #1: N/A

Reviewer #2: Yes

4. Have the authors made all data underlying the findings in their manuscript fully available?

Reviewer #1: (No Response)

Reviewer #2: Yes

5. Is the manuscript presented in an intelligible fashion and written in standard English?

Reviewer #1: (No Response)

Reviewer #2: Yes

6. Review Comments to the Author

Reviewer #1: Thanks so much for the revision, and many of the problems are addressed. But, several problems are raised by the revision, which should also be addressed.

1.It is reasonable that refining an abstract can exclude the interference of too long text and irrelevant text. How to refine the abstract should be the focus of this paper. The extraction method is based on the rules of extraction; the innovation and effectiveness of its method should be evaluated and compared with the abstract extraction method based on deep learning.

2.It makes sense to use character matching because there are special characters and terms, but it should be supported by ablation experiments in the evaluation.

3.The authors say the contribution of modeling is optimizing the computational bias of similarity by introducing reconstruction and scoring sections, but there are no ablation experiments to support it.

4.For the comparison based on the related document dataset, only giving accuracy is not enough. The precision and F1 score should also be given.

Reviewer #2: The authors have responded to reviewer comments and made revisions to address the concerns raised.

Readers without a background in the field may find it difficult to understand the nature of patent documents and multidimensional fusion strategies. For the benefit of the reader, it would be helpful to smooth out several sentences and make the proposed methods and experiments reproducible.

7. PLOS authors have the option to publish the peer review history of their article (what does this mean?). If published, this will include your full peer review and any attached files.

Reviewer #1: No

Reviewer #2: **Yes: **Suan Lee

---

## [Author Response · Author response to Decision Letter 1]

11 Sep 2023

Dear academic editor and reviewers:

Thanks for the careful review of our revised manuscript. Your recognition of our revised content and the proposed new suggestions are important for the further refinement of our manuscript. We have revised the manuscript carefully, and please find the following detailed reply and actions to your comments and suggestions one by one:

Reviewer #1: Thanks so much for the revision, and many of the problems are addressed. But, several problems are raised by the revision, which should also be addressed.

Reply:Thanks for your recognition of our modification work. According to your precious suggestions for the revised manuscript, we have revised the manuscript again carefully, the detailed reply and actions are listed below:

1.It is reasonable that refining an abstract can exclude the interference of too long text and irrelevant text. How to refine the abstract should be the focus of this paper. The extraction method is based on the rules of extraction; the innovation and effectiveness of its method should be evaluated and compared with the abstract extraction method based on deep learning.

Reply:Thank you for pointing this out. The abstract is indeed the most important part of an article, which can make people intuitively understand the research topics and innovation points of this article. In our study, the refinement of the key content of the patent text is achieved through word segmentation, screening, reconstruction and scoring in text preprocessing. For our method, the input is a preprocessed character or word vector, while for the deep learning model, the input is the original patent text. Although the ultimate goal is to assess the similarity of patents, the realization process does have certain differences. As you mentioned, the extraction method based on deep learning is relatively abstract, making it difficult to visually observe the process and results of deep learning models extracting patent text features. Therefore, it is not possible to make a simple comparison between the content extraction results of the deep learning models and ours. Moreover, due to the significant differences in the similarity calculation methods between the deep learning models and ours, it is difficult to determine the impact of these two extraction results on the final similarity calculation. However, this part does give us a new idea for follow-up research, which can visualize the feature extraction process and the final category determination mechanism of the deep learning models to achieve the comparison of the content refining effect and the impact on the subsequent evaluation results.

Action:We have added a quantitative description of the experimental results of our method and other methods on different datasets in the abstract section of the article. And in the methodology and discussion section of the article, a more detailed discussion and effectiveness verification were conducted on the feature extraction method proposed in this article.

2.It makes sense to use character matching because there are special characters and terms, but it should be supported by ablation experiments in the evaluation.

Reply:Thanks for this comment, we agree with your perspective. 

Action:We have added corresponding ablation experiment to better support our viewpoint.

3.The authors say the contribution of modeling is optimizing the computational bias of similarity by introducing reconstruction and scoring sections, but there are no ablation experiments to support it.

Reply:Thanks for this comment. The ablation experiment should be added to demonstrate the effectiveness of these two sections.

Action:We have conducted a comprehensive ablation experiment on the two proposed improvements (reconstruction and scoring) and analyzed the experimental results to support our viewpoint.

4.For the comparison based on the related document dataset, only giving accuracy is not enough. The precision and F1 score should also be given.

Reply:Thanks for this suggestion, we accept it thoroughly.

Action:We have re-evaluated the test results of each model and added three new indicators: precision, recall, and F1 score for a more comprehensive evaluation.

Reviewer #2: The authors have responded to reviewer comments and made revisions to address the concerns raised.

Reply:Thanks for the careful review of our revised manuscript. We sincerely appreciate your recognition of our modification work. We have carefully considered your concerns and further improved the manuscript.

1.Readers without a background in the field may find it difficult to understand the nature of patent documents and multidimensional fusion strategies. For the benefit of the reader, it would be helpful to smooth out several sentences and make the proposed methods and experiments reproducible.

Reply:Thanks for this suggestion, this is indeed a problem that deserves our consideration and resolution.

Action: We have carefully reviewed and refined the language throughout our paper, provided a more accurate description of the proposed method and supplemented with corresponding ablation experiments to further verify the effectiveness of our method.

---

## [Decision Letter · Decision Letter 2]

5 Oct 2023

A multi-dimensional fusion strategy similarity measure method for patent application technology disclosure document

PONE-D-23-12263R2

Dear Dr. Li,

We’re pleased to inform you that your manuscript has been judged scientifically suitable for publication and will be formally accepted for publication once it meets all outstanding technical requirements.

Kind regards,

Anas Bilal, Ph.D.

Academic Editor

PLOS ONE

Reviewers' comments:

Reviewer's Responses to Questions

**Comments to the Author**

1. If the authors have adequately addressed your comments raised in a previous round of review and you feel that this manuscript is now acceptable for publication, you may indicate that here to bypass the “Comments to the Author” section, enter your conflict of interest statement in the “Confidential to Editor” section, and submit your "Accept" recommendation.

Reviewer #1: All comments have been addressed

Reviewer #2: All comments have been addressed

2. Is the manuscript technically sound, and do the data support the conclusions?

Reviewer #1: Yes

Reviewer #2: Yes

3. Has the statistical analysis been performed appropriately and rigorously? 

Reviewer #1: (No Response)

Reviewer #2: Yes

4. Have the authors made all data underlying the findings in their manuscript fully available?

Reviewer #1: (No Response)

Reviewer #2: Yes

5. Is the manuscript presented in an intelligible fashion and written in standard English?

Reviewer #1: (No Response)

Reviewer #2: Yes

6. Review Comments to the Author

Reviewer #1: Thanks for the revision, and all of my questions have been addressed. It is better to release the source code of this work.

Reviewer #2: The authors faithfully revised my comments, analyzed and presented the experimental results technically valid and appropriately.

7. PLOS authors have the option to publish the peer review history of their article (what does this mean?). If published, this will include your full peer review and any attached files.

Reviewer #1: No

Reviewer #2: **Yes: **Suan Lee

---

## [Editor Report · Acceptance letter]

9 Oct 2023

PONE-D-23-12263R2 

A multi-dimensional fusion strategy similarity measure method for patent application technology disclosure document 

Dear Dr. Li:

I'm pleased to inform you that your manuscript has been deemed suitable for publication in PLOS ONE. Congratulations! Your manuscript is now with our production department. 

Kind regards, 

on behalf of

Dr. Anas Bilal 

Academic Editor

PLOS ONE